# A Refined Single Cell Landscape of Haematopoiesis in the Mouse Foetal Liver

**DOI:** 10.3390/jdb11020015

**Published:** 2023-03-23

**Authors:** Elena Ceccacci, Emanuela Villa, Fabio Santoro, Saverio Minucci, Christiana Ruhrberg, Alessandro Fantin

**Affiliations:** 1Department of Experimental Oncology, IEO, European Institute of Oncology IRCCS, Via Adamello 16, 20139 Milan, Italy; 2Department of Biosciences, University of Milan, Via G. Celoria 26, 20133 Milan, Italy; 3Department of Oncology and Hemato-Oncology, University of Milan, Via Santa Sofia 9, 20122 Milan, Italy; 4UCL Institute of Ophthalmology, University College London, 11-43 Bath Street, London EC1V 9EL, UK

**Keywords:** foetal liver, haematopoietic development, haemoglobin

## Abstract

During prenatal life, the foetal liver is colonised by several waves of haematopoietic progenitors to act as the main haematopoietic organ. Single cell (sc) RNA-seq has been used to identify foetal liver cell types via their transcriptomic signature and to compare gene expression patterns as haematopoietic development proceeds. To obtain a refined single cell landscape of haematopoiesis in the foetal liver, we have generated a scRNA-seq dataset from a whole mouse E12.5 liver that includes a larger number of cells than prior datasets at this stage and was obtained without cell type preselection to include all liver cell populations. We combined mining of this dataset with that of previously published datasets at other developmental stages to follow transcriptional dynamics as well as the cell cycle state of developing haematopoietic lineages. Our findings corroborate several prior reports on the timing of liver colonisation by haematopoietic progenitors and the emergence of differentiated lineages and provide further molecular characterisation of each cell population. Extending these findings, we demonstrate the existence of a foetal intermediate haemoglobin profile in the mouse, similar to that previously identified in humans, and a previously unidentified population of primitive erythroid cells in the foetal liver.

## 1. Introduction

The liver is a metabolic hub that regulates glucose and lipid metabolism as well as protein and bile synthesis, but it is also an essential site for both blood and immune system development in mammals. Thus, the foetal liver provides a suitable microenvironment for the expansion and maturation of several waves of multipotent haematopoietic progenitors [1,2]. The liver rudiment emerges as a diverticulum from the ventral domain of the embryonic foregut at the early Carnegie Stage (CS) 10 in humans and on embryonic day (E) 8.75 in the mouse. Most of our knowledge of early liver development is derived from the mouse. Thus, we now know that the diverticulum transitions from a monolayer of cuboidal endoderm cells into a pseudostratified multilayer of hepatoblasts to form the liver bud between E9 and E10 in the mouse [3]. The hepatoblasts then serve as bi-potent progenitors to produce the two major epithelial cell types of the adult liver, hepatocytes and cholangiocytes (also known as biliary epithelial cells), to form the foetal liver [3]. Hepatoblast specification and expansion require endothelial cells, which arise adjacent to the mouse liver diverticulum at E9.0 [3] and specialise to form the lining of the hepatic sinusoids as so-called liver sinusoidal endothelial cells (LSECs). In each hepatic acinus, liver sinusoids are sandwiched between cords of hepatocytes to transport blood from the portal veins and hepatic arteries towards draining central veins, with adult LSECs demonstrated to show zone-specific heterogeneity in both mice and humans [4,5,6].

Before the bone marrow is established as the main postnatal haematopoietic organ shortly before birth in mice and at the beginning of the second trimester in humans, the liver bud is colonised by different waves of haematopoietic progenitor cells [1,2]. Studies using the mouse showed that these include erythro-myeloid progenitors (EMPs) and lympho-myeloid progenitors (LMPs), which emerge from the yolk sac endothelium from E8.5 and E9.5 onwards, respectively, as well as haematopoietic stem and progenitor cells (HSPCs) originating from E10.5 onwards in the intra-embryonic aorta-gonad-mesonephros (AGM) region, whereby the latter population includes the progenitors of the first definitive HSCs, termed pre-HSCs [7,8,9,10]. Pulse chase lineage tracing studies in the mouse further suggested that HSCs, via their immediate progeny called multi-potent progenitors (MPPs), do not contribute significantly to the foetal erythroid, megakaryocyte and myeloid lineages until late in gestation [11,12,13]. 

In adult mammals, erythrocyte differentiation starts when megakaryocyte-erythroid progenitors (MEPs) arising from definitive HSCs progressively differentiate into lineage committed erythroid progenitors, erythroid burst forming unit (BFU-E), erythroid colony forming unit (CFU-E), nucleated proerythroblast, basophilic, polychromatophilic and orthochromatic erythroblast stages, followed by enucleation and formation of reticulocytes and then mature erythrocytes [14,15]. As opposed to the formation of enucleated erythrocytes, the first erythroid cells during embryonic development arise from primitive haematopoietic progenitors in the extra-embryonic yolk sac blood islands at around CS 7–8 (16–18.5 dpc) in the human [16] and E7.5–8.0 in the mouse [17]. Studies in the mouse show that these primitive erythroid cells retain their nucleus and circulate into embryonic organs, including into the liver, where they interact with macrophages in erythroblastic islands to undergo enucleation between E12.5 and E14.5 [14]. Further, mouse studies showed that a subsequent, transient definitive wave of yolk sac-derived EMPs produces CD131 (*Csf2rb*)-positive megakaryocyte-erythroid progenitors (MEPs), which seed the foetal liver to provide the main source of erythrocytes and megakaryocytes in the late gestation embryo [12,14,18]. The transient definitive erythroblasts also interact with foetal liver macrophages to expel their nucleus before entering the circulation at the reticulocyte stage [14].

Erythrocytes express large amounts of haemoglobin, whose subtypes change during prenatal development in both mice and humans. Specifically, oxygen affinity decreases from embryonic to foetal to adult haemoglobin, which is thought to accommodate the changing oxygen tension as the embryo develops to maturity [14,19]. All haemoglobins are tetramers composed of two α-like and two β-like globin chains [19]. This globin nomenclature derives from the two α and β chains that are present in the main adult human form. In adult humans, the most common form is α2β2, in which the two α and β subunits are encoded by the *HBA1* or *HBA2* and *HBB* genes, respectively; the rarer tetramer α2δ2 is instead encoded by *HBA1* or *HBA2* and *HBD*. The human embryonic haemoglobins include the following tetramers: Gower-1 ζ2ε2 (*HBZ*, *HBE1*), Gower-2 α2ε2 (*HBA1/2*, *HBE1*), Portland-1 ζ2γ2 (*HBZ*, *HBG1* or *HBG2*) and Portland-2 ζ2β2 (*HBZ*, *HBB*) [20]. The human foetal haemoglobins (HbF) are termed α2γ2 (encoded by *HBA1* or *HBA2* and *HBG1* or *HBG2*) [20]; these haemoglobins are thought to be a specific feature of anthropoid primates [14]. In the mouse, embryonic β-like globins (εy and βH1) are thought to be restricted to primitive erythrocytes, whereas adult β globin chains are already present in foetal liver-derived erythrocytes [21]. Whether transient definitive erythrocytes in the mouse foetal liver have unique haemoglobin profiles that differ from those of primitive and definitive erythrocytes is not completely understood. 

Mice studies further showed that EMPs and HSPCs in the foetal liver also contribute to myeloid cell production. In particular, EMPs generate liver monocytes that differentiate into tissue-resident macrophages in many organs [22]. These liver monocyte-derived macrophages gradually replace the initial pool of tissue macrophages that is derived from earlier yolk sac primitive progenitors, except for the brain, where microglia derived from primitive progenitors persist [7,11,22,23,24]. Other foetal liver myeloid lineages include granulocytes and mast cells [25]. However, a comprehensive single cell transcriptomic profiling of these early myeloid populations has not been described yet.

Given the major contribution to both erythropoiesis and myelopoiesis, investigating the molecular and cellular landscape of early liver development has the promise to increase our understanding of the causes of congenital immunodeficiencies, anaemia and also childhood leukaemia [26]. Towards this aim, several single cell (sc) RNA-seq datasets have been recently generated, which sought to identify mouse foetal liver cell types via their transcriptomic signature and compared gene expression patterns at single cell level across the cell types in the foetal liver (e.g., [27,28,29,30]. However, the haematopoiesis studies of the mouse foetal liver using scRNA-seq to date focussed on the analysis of selected cell subsets, isolated by genetic lineage-tracing [31] or surface phenotyping [28]. Other scRNA-seq datasets were generated without isolation bias, but from only a small number of foetal liver cells [27,29], which is suboptimal for deep phenotyping, or they were comprised of larger cell numbers but analysed mostly hepatocyte development [30]. Here we describe the generation of a scRNA-seq dataset from E12.5 mouse foetal liver, which was obtained without cell preselection and includes a larger number of cells than prior datasets at this stage. We have combined the analysis of our dataset with data mining of other publicly available datasets to provide new insights into early haematopoietic development in the foetal liver and also included an analysis of foetal liver-constituent cell types.

## 2. Materials and Methods

### 2.1. Library Construction with the 10x Genomics Platform

A foetal liver scRNA-seq dataset was generated from an E12.5 C57BL/6J foetal liver. Briefly, a single cell suspension from one E12.5 liver was mechanically and enzymatically homogenised in RPMI1640 with 2.5% foetal bovine serum (FBS, ThermoFisher), 100 μg/mL collagenase/dispase (Roche), 50 μg/mL DNase (Qiagen) and 100 μg/mL heparin (Sigma), followed by purification from debris, dead cells and doublets through fluorescence activated cell sorting (FACS) (Figure 1a). To maximise the yield in droplet encapsulation of single cells, two separate technical replicates of the sample were independently processed. cDNA Droplet-based digital 3′ end scRNA-seq was performed on a Chromium Single-Cell Controller (10X Genomics, Pleasanton, CA, USA) using the Chromium Single Cell 3′ Reagent Kit v2 according to the manufacturer’s instructions. Cells were divided into 2 samples and independently partitioned in Gel Bead-in-EMulsion (GEMs) droplets and lysed, followed by RNA barcoding, reverse transcription and PCR amplification (12–14 cycles). Sequencing-ready scRNA-seq libraries were prepared according to the manufacturer’s instructions, checked and quantified on TapeStation 2200 (Agilent Genomics, Santa Clara, CA, USA) and Qubit 3.0 (Invitrogen, Carlsbad, CA, USA) instruments. Sequencing was performed on a NovaSeq 6000 machine (Illumina, San Diego, CA, USA) using the NextSeq 500/550 High Output v2 kit (75 cycles) at the genomic facility at the European Institute of Oncology (IEO, Milan, Italy). 

### 2.2. scRNA-seq Data Processing

Fastq.gz files were generated from raw Illumina BCL files of the E12.5 liver sequencing reads using 10X Genomics Cell Ranger version 2.1.1 with default parameters. The quality of sequencing reads was evaluated using FastQC v0.11.5 and MultiQC v1.5. Cell Ranger v2.1.1 was then used to align the sequencing reads to the mm10 mouse transcriptome and to quantify the expression of transcripts in each cell. Only confidently mapped reads, non-PCR duplicates, with valid barcodes and UMIs were retained to compute a gene expression matrix containing the number of UMI for every cell and gene. For each of the two samples, cells expressing less than 200 unique genes and genes expressed in less than 3 cells were discarded. The two replicates were then integrated and log-normalised. The resulting integrated sample had 10537 cells (5480 from sample 1 and 5057 from sample 2). 

### 2.3. scRNA-seq Data Analysis

All downstream analyses were implemented using R v4.2.0 and the package Seurat v4.1.1 (Butler et al., 2018; Stuart et al., 2019). Raw data for the E12.5 foetal liver set were deposited in NCBI’s Gene Expression Omnibus (NCBI-GEO) as GSE180050. Other publicly available datasets analysed in this study include whole mouse embryo scRNA-seq at E8.5 (ArrayExpress, E-MTAB-6967); foetal liver scRNA-seq at E9.5, 10.5 and 11.5 (NCBI-GEO, GSE87038); foetal liver scRNA-seq at E11.0, 11.5 and 13.0 (CNCB-NGDC, CRA002445). 

We performed graph-based clustering for all the analysed datasets. Most variable genes across each dataset (i.e., the highly variable genes) were identified based on the highest standardised variance. The procedure is implemented in the FindVariableFeatures function with method = ‘‘vst.’’ A total number of 2000 genes was selected as top variable features and used to perform the PCA dimensionality reduction. A KNN graph based on the Euclidean distance in the first 15 PCs space was constructed using the FindNeighbors function. A cluster resolution of 1.2 for the E12.5 liver dataset was chosen by comparing 10 different resolutions using clustree v 0.5.0 and then manually curated based on marker genes to obtain the final 11 clusters. 

## 3. Results

### 3.1. Major Cell Populations in E12.5 Mouse Liver in a Novel scRNA-seq Dataset 

To define the cellular landscape of early liver haematopoiesis in the context of other developing liver cell lineages, we generated a scRNA-seq dataset of the mouse E12.5 liver using a droplet-based approach that allowed recovery of a hundred times higher number of cells compared to a previous report, which only sequenced ~100 cells at this stage [29]. Thus, two independent single cell cDNA libraries were prepared from the total single cell suspension of E12.5 foetal liver and sequenced in two batches that were pooled to increase the number of cells sequenced (Figure 1a, Appendix A). The data from 10,537 cells passed quality control and underwent graph-based unsupervised Louvain clustering and dimensionality reduction by Uniform Manifold Approximation and Projection (UMAP) (Appendix A). Differentially expressed genes (DEGs, Appendix A) were used to annotate cell clusters and to pool the most similar clusters into a curated representation of E12.5 liver cell populations (Figure 1b–d). These DEGs are listed in detail in the sections below, where we will discuss specific cell populations. Overall, we identified 11 major cell populations (Figure 1b,c). The identity of each cluster was assigned by matching the cluster expression profile to established cell type-specific genes for erythroid cells (e.g., *Hba-x* for EryP, *Rhd* and *Klf1* for Ery1-6), megakaryocytes (Mk, e.g., *Pf4*, *Itga2b* and *Plek*), immune cells (Imm, e.g., *Ptprc* and *Mpo*), endothelial cells (Endo, e.g., *Cldn5* and *Sox18*) and hepatocytes (Hepa, e.g., *Alb*, *Alf* and *Dlk1*) (Figure 1d). As expected, given the red colour appearance of the foetal liver, ~90% of all E12.5 liver cells were erythroid cells (Figure 1c). The Imm, Mk and Ery1-6 clusters formed a single, branched supercluster when UMAP dimensional reduction was applied (Figure 1b), suggesting that these cell types can share a common haematopoietic progenitor lineage from the immune cell cluster. Instead, hepatocytes and endothelial cells formed segregated clusters from each other and the immune cell cluster, consistent with distinct lineage origins.

### 3.2. Identifying Haematopoietic Progenitors in E12.5 Liver 

We investigated the Imm, Mk and Ery1 clusters in more detail to identify subpopulations of cells that contribute to the immune and erythroid cell population in the E12.5 liver (Figure 2a,b). We identified HSPCs via transcripts for *Cd34* and *Cd93* (coding for AA4.1) (Figure 2c and Appendix A) [13]. DEGs for these cells compared to other liver cells were *Sox4*, *Gimap1*, *Hmga2*, *H2afy*, *Marcks* (Figure 2c, Figure 3a and Appendix A). Unsupervised clustering divided these haematopoietic progenitors into three separate subclusters, which were annotated based on the expression of additional key marker genes. All three subclusters expressed the haematopoietic progenitor markers *Flt3, Kit* and *Myb* at different levels (Figure 2c and Appendix A). By comparing DEGs between these three progenitor populations and based on prior knowledge of foetal haematopoietic progenitor populations [1,2], we identified the cluster with the highest *Ly6a* levels as pre-HSCs/HSCs, the cluster with low *Ly6a* levels but similarly high *Myb* and *Kit* levels as EMP/MPPs and the cluster with high *Flt3* and *Myb* and *Kit* levels as LMPs (Figure 2c and Appendix A).

We further found that the pre-HSC/HSC population had the highest levels of *Hlf*, recently shown to mark the developmental pathway to HSCs but not EMPs [32]. The pre-HSCs/HSC population was also enriched in *Cd27*, *Mycn*, *Hoxa3*, *Hoxa7*, *Hoxa9*, *Rbp1*, *Hacd4*, *Cdkn1c* and *Mecom* (Figure 3b and Appendix A). However, this cell population did not yet express *Slamf1* (encoding CD150) (Figure 3b), a widely accepted marker used to label foetal HSCs at late gestation [13]. The LMP population included *Ccr9* and *Il7r*-positive cells, as recently shown [28] and specifically expressed *Igll1*, *Klrd1* and *Pax5*. The LMP population was also enriched in transcripts for *Ccl3*, *Ccl4*, *Pld4*, *Lsp1*, *Plac8*, *Ckb* and *Mndal* but lacked transcripts for the HSC and MPP markers *Hlf* and *Cd48*, respectively (Figure 3b and Appendix A). The EMP/MMP population was enriched for *Fcgr3* transcripts (encoding CD16) and *Cd48* (Figure 3b and Appendix A), which are known EMP and MPP markers, respectively, but not HSC markers [12]. The distinctive gene signature of the EMP/MMP population also included *Mpo*, *Calr*, *Ccl9*, *Ap3s1*, *Cpa3*, *Anxa3* and *Ctsg* (Figure 3b and Appendix A). The yolk sac EMP marker *Csf1r* [11,23] was expressed at low levels in the EMP/MMP population (Figure 2c), but less prominently than in monocytes and Kupffer cells (see below; Figure 2c). By contrast, *Csf1r* transcripts were hardly detected in pre-HSCs/HSCs (Figure 2c). Therefore, *Csf1r* expression in the presence of progenitor markers can be used to distinguish EMPs or other MPPs from pre-HSCs/HSCs at this developmental stage.

### 3.3. Molecular and Cellular Landscape of Megakaryocyte, Erythroid and Myeloid Lineages in E12.5 Liver

UMAP dimensionality reduction showed that the EMP/MPP cluster formed a differentiation continuum with two neighbouring cell clusters, the megakaryocyte-erythroid progenitors (MEPs) and granulocyte-monocyte progenitors (GMPs) (Figure 2a). Marker and DEG analysis of all the subclusters within the Imm, Mk and Ery1 populations suggested that MEP and GMP underwent commitment already at E12.5 along the erythro-megakaryocytic and the myeloid lineages, respectively, as described below.

MEPs could be distinguished from other HSPC populations by higher transcript levels for *Myb* and *Kit*, together with the erythroid markers *Gata1* and *Klf1* and the erythroid-megakaryocytic marker *Smim1*. The top DEGs in MEPs were *Muc13* and *Car1*. MEPs formed lineage trajectories towards erythroid progenitors expressing *Klf1*, *Rhd* and *Tfrc*, identified as BFU-E (see Figure 4), and megakaryoblasts and megakaryocytes expressing *Pf4* and *Plek* (Figure 2c, Figure 3c and Appendix A). The top DEGs in erythroid progenitors were *Pla2g12a*, *Asns* and *Rexo2*, whereas the top DEGs in megakaryoblasts and megakaryocytes were *F2rl2*, *Rab27b*, *Gp1bb*, *Pbx1*, *Gp9* and *Rap1b*. Megakaryoblasts were also enriched in transcripts for *F2r*, *Pbx1*, *Lat* and *Clec1b*, whereas megakaryocytes were enriched in transcripts for *Myl9*, *Gp1ba*, *Itga2b*, *Cd226*, *Mest* and *Hist1h2bc* (Figure 3c and Appendix A).

GMPs could be distinguished from other HSPC populations by their enrichment in transcripts for *Hdc, Perp* and *Fcgr3* (Figure 3c and Appendix A). The GMP population formed lineage trajectories towards granulocyte progenitors enriched in *Ms4a3*, *Mpo* and *Fcgr2b*, with the top DEGs being *Cd63* and *Elane* (Figure 2c, Figure 3c and Appendix A). Granulocyte progenitors then led to neutrophils/granulocytes, which were enriched in transcripts for *Itgam* encoding CD11b, *S100a8*, *S100a9* and *Ly6g*; other top DEGs were *Retnlg*, *Camp*, *Ngp*, *Ltf*, *Lcn2*, *Ifitm6* and *Stfa1* (Figure 2c, Figure 3c and Appendix A). A parallel lineage trajectory led from GMPs to Kupffer cells/tissue macrophages, which were enriched in transcripts for *Aif1* encoding IBA1, *Cx3cr1*, *Mertk* and *Clec4f*, and to monocytes, which were enriched in transcripts for *Ccr2*, *Ly6c2 and Pld4* (Figure 2c, Figure 3c and Appendix A). Monocytes and granulocyte progenitors shared high levels of *Lgals3* and *Fcnb* transcripts with each other and further shared *Lyz2* and *Hp* transcripts with neutrophils (Figure 2c and Appendix A). *Csf1r* transcripts were high in Kupffer cells/tissue macrophages and even higher in monocytes, but not detected in differentiated neutrophils (Figure 2c). *Csf1r* expression can therefore be used as a distinguishing marker for the macrophage/monocytic branch within the myeloid lineage. Cells enriched in mast cell markers *Cpa3* and *Runx1* transcripts [33] appeared to bud from the GMP cluster (Figure 3c) and may represent early foetal mast cells. 

### 3.4. Cell Cycle Analysis of Haematopoietic Lineages in the E12.5 Liver

The CellCycleScoring prediction algorithm in Seurat together with expression analysis for the S marker *Pcna* [34] and S/G2 marker *Mki67* [35] reported that cell number expansion mainly occurs in MEPs as well as downstream of MEPs in the megakaryoblasts and BFU-E of the megakaryocyte/erythroid branch as well as in granulocyte progenitors of the myeloid branch (Figure 3d,e). Cell number expansion was more moderate in HSPCs at E12.5, with pre-HSCs/HSCs appearing less proliferative than other progenitors at this stage (Figure 3d,e). Liver megakaryocytes appeared to have already terminally differentiated at E12.5, because this cluster lacked cells entering the G2/M phase, with less than 10% of cells in this cluster in the S phase (Figure 3d,e). Within the myeloid lineage, neutrophils also appeared already terminally differentiated, whereas the majority of Kupffer cells and especially monocytes were either in the S or G2/M phase (Figure 3d,e). 

### 3.5. Molecular and Cellular Landscape of Erythropoiesis in E12.5 Liver

Mouse foetal liver erythroid cells were identified by their expression of haemoglobin genes, the erythroid-specific transcription factor *Klf1* and the erythroid membrane marker *Rhd* (Figure 1d). We first examined the expression pattern of the nuclear long noncoding RNA (lncRNA) *Malat1* as a nucleus marker [36]. As *Malat1* was present in all cells of the erythroid clusters identified in the E12.5 liver, they all contained a nucleus (Appendix A), consistent with an origin from yolk sac-born primitive erythrocytes circulating through the liver (EryP cluster) and from liver-resident transient-definitive erythroid progenitors and erythroblasts (Ery1–6 clusters). The total number of expressed genes decreased gradually from Ery3 to Ery5, resulting in a drastic reduction of transcriptomic complexity in both Ery6 and EryP (Appendix A), agreeing with an erythroid maturation program that includes enhanced chromatin condensation and consequent transcription silencing [15]. 

CellCycleScoring prediction together with *Pcna* and *Mki67* expression analysis suggested that no cells in the EryP cluster were in the S phase and fewer than 10% in G2/M (Figure 4a,b), consistent with them being terminally differentiated erythrocytes. The Ery2-5 clusters, instead, contained the most proliferative cells in the whole liver (Figure 4a,b), suggesting rapid expansion of erythroid progenitors or erythroblasts in the E12.5 liver. Cells in the Ery6 cluster were mostly in the G1 phase similar to the EryP cluster, with no cells in the S phase and 20% in G2/M (Figure 4a,b) and therefore likely represent orthochromatic erythroblasts that had ceased to divide [37]

Similarly to what was recently reported by flow cytometry [12], gene expression analysis showed that the Ery1 cell population contained transcripts for *Myb* and *Kit* but not *Cd24a*, and thus likely corresponds to BFU-E progenitors. The Ery2 cell population expressed higher levels of *Cd24a* compared to Ery1, but remained haemoglobin-negative, and thus likely corresponds to CFU-E progenitors. Transcripts for *Cd24a*, *Tfrc* encoding CD71 and haemoglobin genes increased from the Ery3 to the Ery4 and then Ery5 cell populations (Figure 4a); these populations therefore likely correspond to sequential erythroid stages proerythroblasts, basophilic erythroblasts and polychromatophilic erythroblasts. Orthochromatic erythroblasts (Ery6 cluster) and primitive erythrocytes (EryP) both expressed the mature erythrocyte marker *Bpgm*, but orthochromatic erythroblasts contained transcripts for *Cd24a* and *Tfrc*, whereas these were barely detectable in primitive erythrocytes (Figure 4a,c). *Bpgm* together with *Cd24a* and *Tfrc* expression could therefore be used as distinguishing markers between orthochromatic erythroblasts and primitive erythrocytes. 

### 3.6. Haemoglobin Gene Expression Profiles in the E12.5 Liver

In the E12.5 liver, transcripts for the embryonic α-like globin *Hba-x* (ζ) and β-like globin *Hbb-y* (εy) were restricted to the EryP cluster, which also expressed the α globins *Hba-a1* and *Hba-a2* (α) as well as the β-like globin *Hbb-bh1* (βH1), but low levels of adult β globins *Hbb-bs* and *Hbb-bt* (Figure 4a,d; note that *Hbb-bs* and *Hbb-bt* are specifically present in C57BL/6 mice, whereas BALB/c and 129Sv mice have a different haplotype that includes *Hbb-b1* and *Hbb-b2*). To corroborate that the EryP cluster represents primitive erythrocytes, we also analysed scRNA-seq data from whole embryos with yolk sacs at E8.5 (ArrayExpress, E-MTAB-6967) [38], when primitive erythrocytes arise in the yolk sac [17]. EryP at E8.5 were identified by *Klf1* and *Rhd* (Appendix A) and were nucleated (*Malat1*-positive; Appendix A). Similar to E12.5, EryP at E8.5 had high levels of *Hba-x* and the *Hba-a1* and *Hba-a2* (α) as well as *Hbb-y* and *Hbb-bh1* (βH1), but barely detectable levels of adult β globin *Hbb-bs* and *Hbb-bt* (Appendix A). By contrast to EryP cells at E12.5, however, EryP at E8.5 still expressed low transcript levels for the mature erythrocyte marker *Bpgm* (Appendix A). The adult β globin transcripts *Hbb-bs* and *Hbb-bt* were highly expressed in the erythroid clusters Ery3, Ery4, Ery5, Ery6 of E12.5 liver alongside the α globins *Hba-a1* and *Hba-a2*, (Figure 4a,d). These clusters also retained *Hbb-bh1* expression, although at lower levels than in primitive erythrocytes (EryP) (Figure 4a,d). Foetal liver erythropoiesis can therefore be distinguished from primitive erythrocytes by the expression of both adult β globin genes and the embryonic β-like globin *Hbb-bh1*.

### 3.7. Onset of Foetal Liver Haematopoiesis

To better understand the onset of haematopoiesis in the mouse foetal liver, we performed a time-course single cell analysis with published scRNA-seq dataset (NCBI-GEO, GSE87038) of mouse foetal liver cells at E9.5, E10.5 and E11.5 [27]. 

In the E9.5 mouse liver, we identified *Afp*-positive hepatocytes and *Col3a1*-positive mesenchymal cells, but *Klf1*-positive erythroid and *Cx3cr1*-positive myeloid cells could not be detected at this stage, with just one cell out of the total 92 cells expressing the EMP markers *Kit*, *Itga2b*, *Cd34* and *Csf1r* (Appendix A). At E10.5, we identified many haematopoietic progenitors expressing the EMP markers *Cd34*, *Csf1r*, *Mpo*, *Fcgr3* and *Ptprc*, and they formed a lineage continuum with myeloid cells expressing *Cx3cr1* and *Csf1r* and megakaryoblasts expressing *Itga2b* (Appendix A). Haematopoietic progenitors (including EMPs) began to express transcripts for *Myb, Fcgr2b* (coding for CD32) and the MPP marker *Cd48* by E11.5 while downregulating transcripts for *Kit* and *Itga2b* (Appendix A).

Coincident with liver homing by EMPs from E10.5 onwards, *Klf1*-expressing erythroid clusters could be detected in the foetal liver from E10.5 onwards and increasingly at E11.5, when they were clearly distinguishable from *Hba-x* positive primitive erythrocytes (EryP) (Appendix A). The liver-derived erythroid cells at both E10.5 and E11.5 included *Myb*- and *Kit*-positive cells (Ery1; Appendix A), as well as *Cd24a*-expressing cells that did not express any haemoglobin genes (Ery2; Appendix A), indicating that they are BFU-E and CFU-E, respectively. A separate cluster of erythroid cells expressing *Tfrc* and low levels of *Hbb-bh1*, representing the first foetal erythroblasts, could be identified in the E11.5 liver (Ery3; Appendix A), further validating the existence of a foetal intermediate haemoglobin profile. 

We validated our observations by analysing another publicly available dataset (CNCB-NGDC, CRA002445) that includes 14,597, 27,998 and 16,592 foetal liver cells at E11.0, E11.5 and E13.0, respectively [30]. At all three stages, haematopoietic progenitors expressing *Kit*, *Myb*, *Cd34*, *Mpo*, and *Ptprc* formed a lineage continuum with MEPs expressing *Muc13*. At both E11.0 and E11.5, liver MEPs expressed *Csf2rb*, a marker for yolk sac MEPs destined to colonise the foetal liver [18]. At these stages, *Csf2rb* expression was also observed in HSPCs (except LMPs) as well as in myeloid cells (Figure 5a–c and Figure 6a–c). At E12.5 and E13.0, *Csf2rb* transcripts remained abundant in myeloid lineages and, especially, in mast cells, but had been lost from liver MEPs (Figure 3c and Figure 7a–c). MEPs further branched into *F2r*-positive megakaryoblasts followed by *Myl9*-expressing megakaryocytes (Figure 5a–c, Figure 6a–c and Figure 7a–c) and *Klf1*- and *Myb*-positive but *Cd24a*-negative BFU-E erythroid progenitors, which led via *Cd24a*-positive CFU-E progenitors to the first erythroblasts with *Hbb-bh1* and *Hbb-bs* haemoglobin expression (Figure 5a–d, Figure 6a–d and Figure 7a–d).

At all three stages, we identified primitive erythroid cells with reduced transcriptomic complexity that were enriched in embryonic but not adult globins and likely are yolk sac-born primitive erythrocytes circulating through the liver (EryP cluster, Figure 5d, Figure 6d and Figure 7d; compare to Figure 4 and Appendix A). Additionally, at E11.0 and E11.5, a small population of erythroid cells expressed a higher number of genes, and they were enriched in embryonic but not adult haemoglobin genes; further, their cluster proximity in the UMAP suggested that their transcriptomic signature is more similar to that of liver-derived erythroblasts than yolk sac-derived primitive erythrocytes (liver EryP; Figure 5d and Figure 6d). These erythroid cells could not be identified in our E12.5 dataset (Figure 4c,d), and very few were detected in the E13.0 dataset (Figure 7d), suggesting that they might be the product of short-lived primitive erythroid progenitors that had homed to the foetal liver before E11.0. 

Haematopoietic progenitors also formed a lineage continuum with myeloid cells, including monocytes expressing *Ccr2*, Kupffer cells expressing *Mertk* and granulocyte progenitors expressing *Ms4a3* (Figure 5a–c, Figure 6a–c and Figure 7a–c). Granulocyte progenitors were already present at E11.0, but the first *Ly6g*-expressing neutrophils/granulocytes appeared only at E11.5 (Figure 5b,c and Figure 6b,c). From E11.5 onwards, mast cells formed a cluster distinct from GMPs, and at E13.0, the mast cell cluster had become larger and appeared phenotypically closer to MEPs when compared to earlier stages (Figure 5b,c, Figure 6b,c and Figure 7b,c). We identified two distinct subsets of tissue macrophages in the E13.0 dataset. One cluster with high *Mertk* and *Csf1r* levels was already present at earlier stages and represented Kupffer cells (KC; Figure 7a–c), which are of yolk sac origin [7]. A second subset of tissue macrophages expressing *Csf1r*, *Adgre1* and *Cx3cr1*, but not the monocyte marker *Ccr2* or the mature KC marker *Mertk*, was present within a cluster that also contained *Ccr2*-positive but *Adgre1*-negative monocytes (Mo/TM; Figure 7a–c). This subset likely represents liver monocyte-derived tissue macrophages that will gradually replace the initial pool of yolk sac-derived Kupffer cells during late gestation [7].

*Ly6a*-positive HSCs or pre-HSCs were rare in E12.5 or E13.0 mouse foetal liver (Figure 2c and Figure 7c), and could not be identified at earlier stages (Figure 5c and Figure 6c). Further, HSCs expressing *Slamf1* (encoding CD150) were not detected at any time point up to E13.0 (Figure 5c, Figure 6c and Figure 7c). By contrast, rare *Flt3*-, *Hlf*- and *Mecom*-positive MPPs as well as *Il7r*- and *Klrd1*-positive LMPs could be identified at all stages (Figure 5c, Figure 6c and Figure 7c). Prior to E12.5, foetal liver haematopoiesis is therefore predominantly driven by transient definitive progenitors first appearing in the mouse liver at E10.5, rather than definitive progenitors.

### 3.8. Non-Haematopoietic Cells in the E12.5 Liver

In addition to haematopoietic cells, the foetal liver contains structural cell types crucial for liver growth and function. These cells segregated apart from the blood and immune cells in our scRNA-seq dataset of the E12.5 liver into three main clusters. 

One of these cell clusters had high levels of hepatoblast markers such as *Afp* and *Alb*, but lacked the mature hepatocyte marker *Epcam* (Hepa cluster; Figure 1b–d). Unsupervised subclustering identified two separated subclusters (Appendix A). One subcluster had higher levels of hepatoblast markers (*Afp*, *Alb*, *Hnf4a, Krt8*) than the other and also upregulated the hepatocyte differentiation markers (*Tbx3*, *Cebpa* and *Prox1*; Appendix A). Based on these known hepatocyte markers (Yang et al., 2017), cells in this cluster may be hepatoblasts undergoing hepatocyte differentiation. The other subcluster had reduced expression of hepatoblast markers (*Afp*, *Alb*) and lacked transcripts for other hepatoblast markers (*Hnf4a, Krt8*) but expressed *Spp1* (Appendix A). *As Spp1* is a marker for cholangiocytes [39], cells in this cluster may be hepatoblasts undergoing cholangiocyte differentiation. Analysis of differentially expressed genes (DEGs) showed that the presumed cholangiocyte precursors were enriched in *Cct4*, *Cct7*, *Eif4a1*, *Ptma* and *Actb*, whereas *Gpc3*, *Hpx*, *Apoh*, *Serpina1c*, *Rrbp1*, *Igf2*, *Fgg*, *Meg3* and *Elovl2* were underrepresented when compared to the presumed hepatoblasts (Appendix A). Another separate cluster was enriched in *Pdgfrb, Acta2, Dcn, Cxcl12, Col3a1* and *Des* (Figure 2a–c and Figure 3f). with other DEGs being *Nnat, Col1a2, Ptn, Sparc, Meg3* and *H19* (Appendix A). Based on prior knowledge (Gordillo et al., 2015), this cluster may be derived from hepatic stellate cell precursors.

A third cluster, termed Endo, contained transcripts for the endothelial markers *Cldn5* and *Sox18* together with transcripts typical of liver sinusoidal endothelial cells (LSECs), such as *Lyve1* and *Plvap* (Figure 1b–d). Unsupervised subclustering revealed the presence of 2 distinct cell populations, termed here LSEC1 and LSEC2 (Appendix A). Both subclusters expressed low levels of *Tm4sf1* and *Clec14a* transcripts (Appendix A), previously identified as periportal LSEC markers in adult human liver [4]. The LSEC1 subcluster appeared enriched for *Oit3* and *Cldn5* (Appendix A), recently reported to be increased in LSECs in central vein proximity [4]. The LSEC1 subcluster was also enriched for *Eng* (Appendix A), which is upregulated in endothelial cells from large calibre vessels in the adult liver, such as the central veins [4]. Further, the LSEC1 subcluster was enriched for *Ephb4* (Appendix A), an Eph receptor selectively expressed on vein endothelial cells [40]. Other DEGs for LSEC1 include *Asb4*, *Ppp1r14b*, *Cnbp*, *Col4a2*, *Plk2*, *Polr2e*, *Tubb6, Plpp3*, *Lpar6*) (Appendix A). By contrast, the top DEG for LSEC2 was *Bex2* (Appendix A), although this cluster lacked a zonation-specific gene expression signature typical of adult liver endothelial cells [4]. 

## 4. Discussion

Defining the mouse embryonic liver environment at single-cell resolution using scRNA-seq informs whether and how the mouse provides a suitable model to understand the molecular bases of human liver development, function and disease, including congenital immunodeficiencies, anaemia and also childhood leukaemia [26]. Here we addressed limitations in prior studies that either performed a preselection step to enrich the dataset only for specific cell types [28,31] or included only a small number of all foetal liver cell types [27,29] and extended the analysis of a study that obtained the transcriptomes of a larger number of liver cells but analysed mostly hepatocyte development [30]. 

Consistent with current knowledge generated by lineage tracing in combination with histology and flow cytometry [1,2], we found that haematopoietic cells were rare in the foetal liver at E9.5 but consistently present in this organ from E10.5 onwards (Appendix A). Further, our data agree with erythroid cells derived from liver-resident, transient-definitive progenitors becoming the main cellular component of the foetal liver by E12.5 (Figure 1). We observed that transient-definitive erythroblasts generated in the foetal liver retained expression of the embryonic β-like globin *Hbb-bh1* (βH1) together with α globins *Hba-a1* and *Hba-a2*, although at lower levels than in primitive erythrocytes (Figure 4). This is interesting because it was previously suggested that intermediate foetal haemoglobin composed of α globins and a foetal β-like globin is a specific feature of anthropoid primates, with βH1 globin expressed only by primitive erythrocytes in the mouse [14]. Our observations instead agree with recent descriptions of low βH1 transcript levels in the mouse foetal liver [12,41]. These prior studies used in situ hybridisation and flow cytometry analyses to detect βH1 transcripts, but these techniques were not combined with markers suitable to clearly distinguish liver transient-definitive erythroblasts from primitive erythrocytes that circulate through the foetal liver, where both types of erythroid cells become enucleated following interaction with macrophages in the erythroblastic islands. These limitations are readily addressed by scRNA-seq data, because the two types of erythroid cells segregated into two distinct clusters (Figure 4, Figure 6 and Figure 7). Our scRNA-seq analysis also identified a short-lived population of erythroid cells in E11.0 and 11.5 liver; these were phenotypically similar to liver-derived erythroblasts and clustered separately from primitive erythrocytes, despite being enriched in primitive but not definitive haemoglobin genes (Figure 5 and Figure 6). These cells might be a hitherto unidentified progeny of a late primitive erythroid progenitor or, alternatively, of an early EMP-like progenitor that colonises the early foetal liver but are rapidly replaced when transient definitive erythropoiesis takes hold.

Erythroid, megakaryocytic and myeloid cells each emerged in a separate, continuous trajectory from the HSPC cluster in all datasets analysed between E10.5 and E13.0 (Figure 1, Figure 2, Figure 4, Figure 5, Figure 6, Figure 7 and Appendix A). This observation suggests that these cell types share a common haematopoietic progenitor in the foetal liver up to E13.0. HSPCs shared expression of *Cd34*, *Cd93* (coding for AA4.1), *Hmga2*, *Flt3*, *Kit* and *Myb*. By contrast, up to E13.0, cells within the HSPC cluster did not contain transcripts for *Slamf1*, encoding CD150 (Figure 2), which is a widely accepted marker for foetal HSCs. Notably, the *Hlf*- and *Ly6a*-expressing preHSC/HSC subset within the HSPC cluster did not appear more proliferative than other progenitors at E12.5 (Figure 3). As HSCs expand in the foetal liver but minimally contribute to foetal haematopoiesis [13], our observations indicate that the E12.5 liver is mostly populated by pre-HSCs or short-term HSCs (ST-HSCs) rather than definitive HSCs, as previously suggested [42]. 

Cell cycle analysis further suggested that haematopoietic cell expansion in the early foetal liver mainly occurs downstream of HSPCs at the level of bipotent or monopotent progenitors, such as MEPs, megakaryoblasts, BFU-E progenitors and granulocyte progenitors (Figure 3). Cell expansion appeared to occur similarly in MEPs as well as megakaryoblasts and BFU-E progenitors in the megakaryocyte/erythroid branch (Figure 3). Instead, different branches of the myeloid lineage showed divergent expansion strategies. Thus, granulocyte progenitors actively proliferated to produce mature neutrophils/granulocytes that had exited the cell cycle (Figure 3). This observation may indicate that these cells expand in the foetal liver before being released into the blood stream. By contrast, differentiated Kupffer cells and monocytes were highly proliferative (Figure 3). It is conceivable that Kupffer cells expand concomitantly with rapid liver expansion to maintain sufficient density, whereas monocytes proliferate to provide enough circulating progenitors for tissue-resident macrophages in peripheral tissues, such as the lung, heart and skin. 

Separate from the haematopoietic cells, the progenitors of hepatocytes and also endothelial cells each formed segregated, small clusters. The small cluster size for these epithelial cell types (Figure 1) was likely explained by cell types being underrepresented following cell isolation, due to them being bound into cell sheets that need dissociating and detaching from the basement membrane, therefore increasing the frequency of cell damage when compared to naturally singular haematopoietic cells. Further, it is conceivable that downstream processing damages adherent endothelial and epithelial cells more readily than haematopoietic cells, because the latter are inherently adapted to fluid sheer stress, which affects cells in suspension for FACS or microfluidic processing. To overcome these technical challenges, future studies might optimise epithelial cell isolation and processing protocols for FACS- or droplet-based scRNA-seq approaches. Alternatively, single nucleus RNA-seq could be used to better characterise endothelial and epithelial cells in the foetal liver.

It was previously reported that hepatoblasts start differentiating into hepatocytes and cholangiocytes at around E13.5 in mice, with cholangiocyte precursors being generated at the ductal plate from a monolayer of hepatoblasts surrounding the portal veins, whereas hepatoblasts located away from portal vein areas differentiate into hepatocytes later on [3]. Our observations instead suggest that the specification of cholangiocyte precursors from the common hepatoblast progenitor already starts at E12.5. The endothelial cell cluster identified in our scRNA-seq of the whole E12.5 mouse foetal liver revealed two distinct cell populations, namely LSEC1 and LSEC2 (Appendix A). LSEC1 appeared polarised towards a central vein but not periportal fate, likely reflecting the remodelling pattern already occurring in the liver bud. LSEC2 instead lacked a zonation-specific phenotype. An absent periportal LSEC phenotype might be explained by incomplete remodelling of the right and left vitelline veins into the portal vein or the need for a switch from foetal to postnatal circulation. 

Other non-haematopoietic cells in the E12.5 foetal liver are mesenchymal in nature, such as the precursors of hepatic stellate cells (Figure 2). Hepatic stellate cells originate from the septum transversum-derived mesothelium during development and localise in the space of Disse in the adult liver, where they constitute the major mesenchymal component to support liver homeostasis; when activated by injury or infection, stellate cells are the major cell type responsible for liver fibrosis. We found that foetal liver stellate cells were highly enriched in *Cxcl12* transcripts, consistent with their role in recruiting CXCR4-expressing haematopoietic progenitors [3].

Owing to its genetic tractability, the mouse is the most widely used mammalian model system to understand human development, including liver morphogenesis and haematopoiesis. Although these developmental processes are generally considered conserved across vertebrates [3,43], there are notable differences between mice and humans, for example in the expression and function of the surface molecules used to immunophenotype haematopoietic progenitors and mature immune cells [16,44]. Accordingly, defining haematopoietic development in the mouse and human foetal liver at the single cell level provides an essential resource for comparing the developmental dynamics that might be affected in congenital or paediatric immune diseases and understanding how these diseases might be modelled in the mouse.

## Figures and Tables

**Figure 1 jdb-11-00015-f001:**
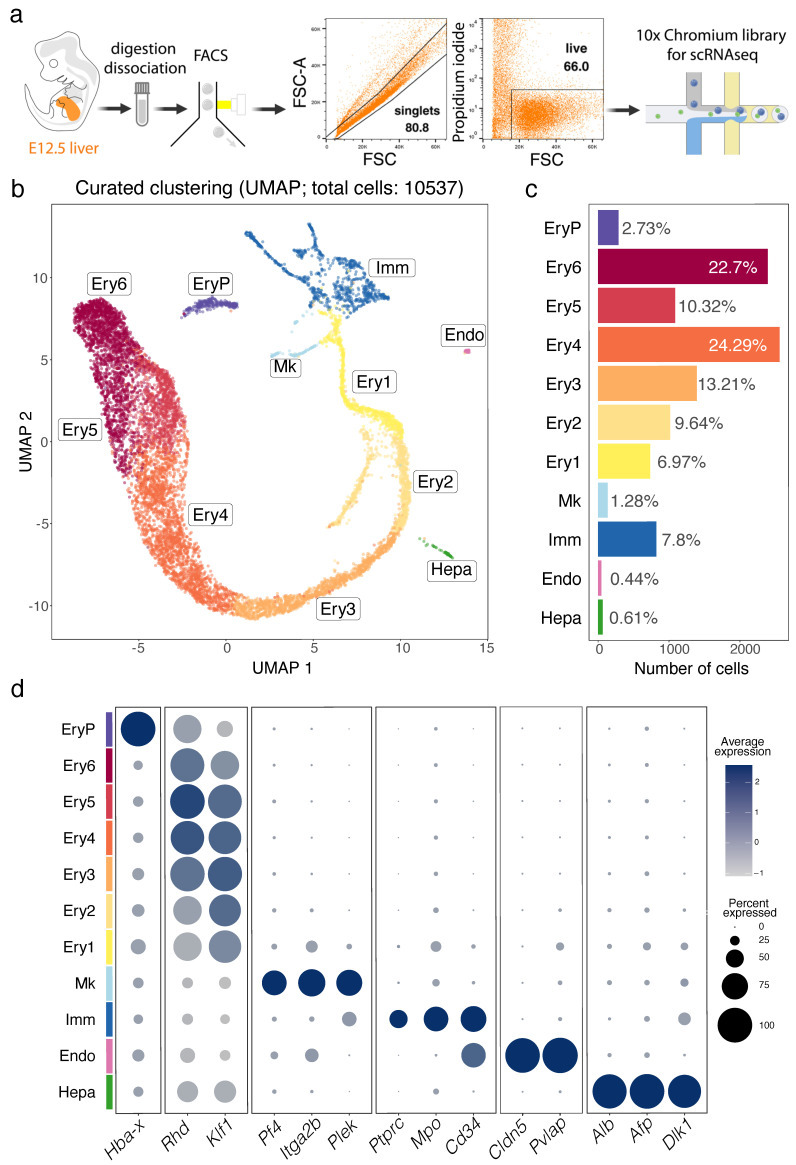
scRNA-seq analysis of the E12.5 mouse liver. (**a**) Gating strategy for E12.5 liver cell isolation prior to single cell library preparation. (**b**–**d**) A UMAP plot shows distinct cell types (**b**), a histogram shows the percentage and number of cells in each cluster (**c**), and a bubble plot shows expression of selected marker genes in each cluster (**d**). Dot size in (**d**) represents the percentage of cells within a cluster in which a marker was detected, dot colour intensity represents the average expression level of that marker. Abbreviations: Endo, endothelial cells; Ery1-6, erythroid cells; EryP, primitive erythrocytes; Hepa, hepatoblasts; Imm, immune cells; Mk, megakaryoblasts.

**Figure 2 jdb-11-00015-f002:**
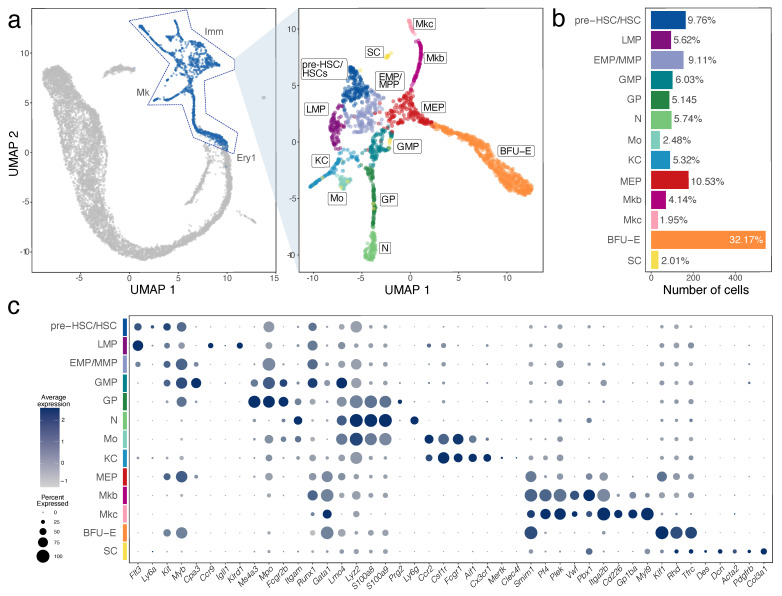
Haematopoietic cell identification in E12.5 mouse liver by scRNA-seq analysis. (**a**) Subset selection followed by UMAP plot visualisation identifies subclusters of distinct haematopoietic cell types. (**b**) A histogram shows the percentage and total number of cells in each subcluster. (**c**) A bubble plot shows expression of selected marker genes in each subcluster; the dot size in represents the percentage of cells within a cell cluster in which a marker was detected, and the dot colour intensity represents the average expression level of that marker. Abbreviations: BFU-E, erythroid burst forming unit; EMP, erythro-myeloid progenitor; GMP, granulocyte-monocyte progenitor; GP, granulocyte progenitor; HSC, haematopoietic stem cell; KC, Kupffer cell; LMP, lympho-myeloid progenitor; MEP, megakaryocyte-erythroid progenitor; Mkb, megakaryoblast; Mkc, megakaryocyte; MMP, multipotent progenitor; Mo, monocyte; N, neutrophil; SC, stellate cell.

**Figure 3 jdb-11-00015-f003:**
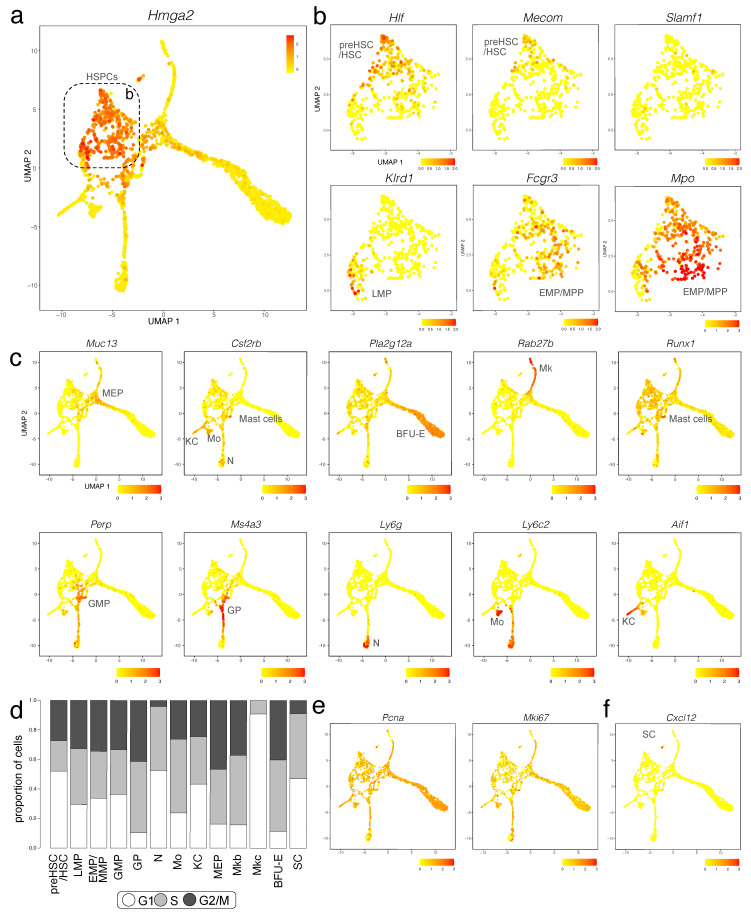
E12.5 mouse liver haematopoietic cells analysed with scRNA-seq data. (**a**,**b**) A UMAP plot of E12.5 mouse liver haematopoietic cells visualises *Hmga2* expression (**a**), which was used to select haematopoietic progenitor subclusters, indicated with a box and shown at higher magnification in (**b**) for expression of the indicated progenitor genes. Each UMAP plot names the cluster(s) expressing the indicated gene. (**c**) UMAP plots E12.5 mouse liver haematopoietic cells visualise expression of the indicated markers for distinct branches of haematopoietic cell differentiation and hepatic stellate cell progenitors. Each UMAP plot names the cluster(s) expressing the indicated gene. (**d**) The CellCycleScoring prediction algorithm identifies the proportion of cells in G1, S, or G2/M cell cycle phases for each haematopoietic cell cluster. (**e**,**f**) UMAP plots visualise expression of *Pcna* and *Mki67* (**e**) and *Cxcl12* (**f**).

**Figure 4 jdb-11-00015-f004:**
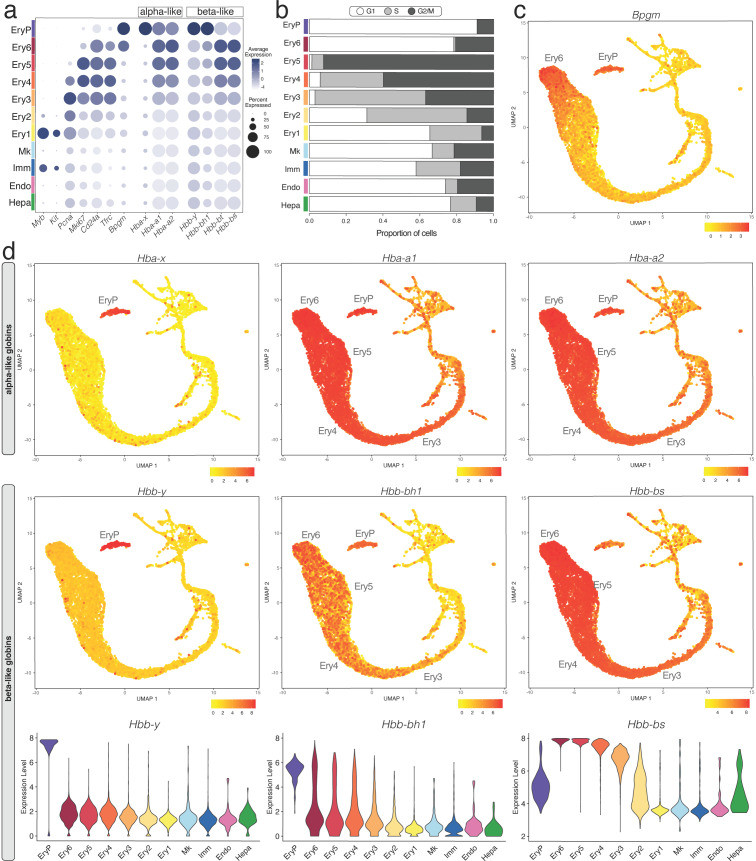
E12.5 mouse liver erythropoiesis analysed with scRNA-seq data. (**a**) A bubble plot shows expression of the selected marker genes in each E12.5 foetal liver cluster; the dot size represents the percentage of cells within a cell cluster in which that marker was detected, and the dot colour intensity represents the average expression level. (**b**) The CellCycleScoring prediction algorithm identifies the proportion of cells in G1, S, or G2/M cell cycle phases for each cluster. (**c**) Expression of the mature erythrocyte marker *Bmpg*. (**d**) UMAP and violin plots visualise the α-like and β-like globin genes expressed in different erythrocyte lineages. In (**c**,**d**), the cluster(s) expressing the indicated gene are named.

**Figure 5 jdb-11-00015-f005:**
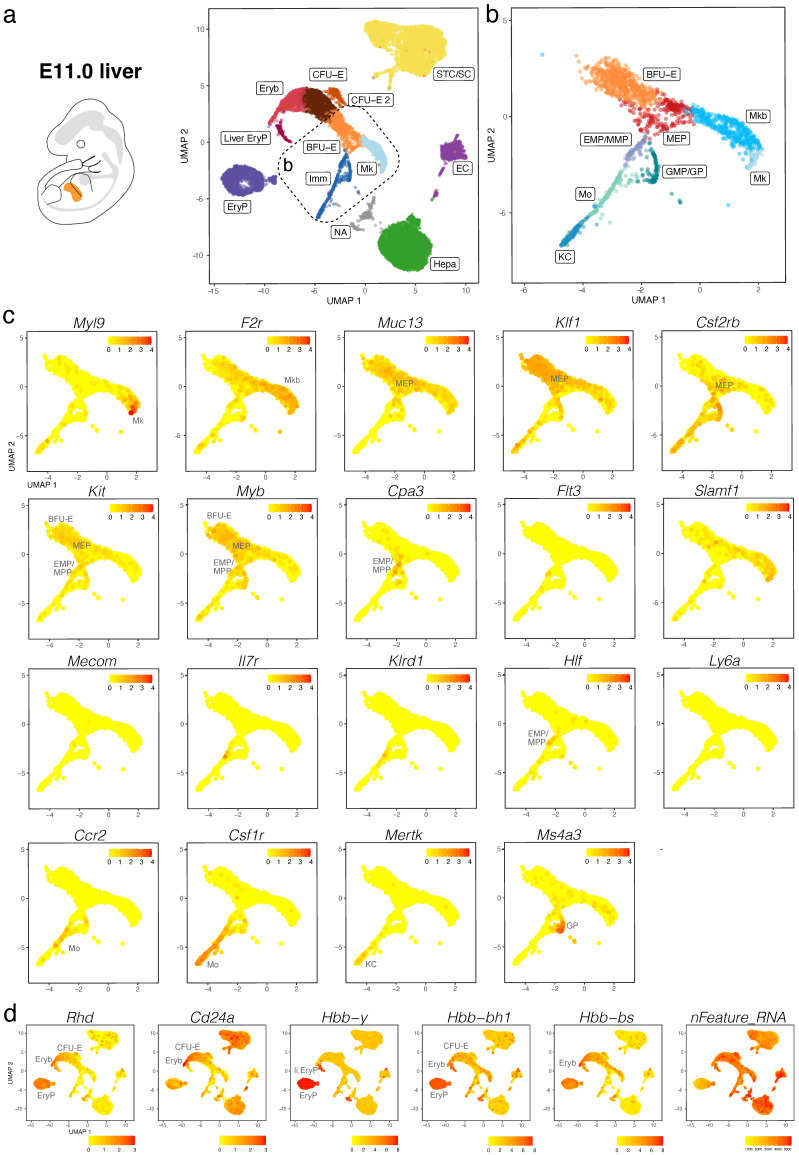
scRNA-seq of the E11.0 mouse liver. (**a**,**b**) The UMAP plot in (**a**) visualises clusters of distinct cell types in the total dataset; the box indicates the Mk, BFU-E and Imm clusters, which were subclustered in (**b**). (**c**,**d**) UMAP plots visualise expression of the indicated genes that serve as markers of distinct branches of haematopoietic cell (**c**) and erythroid differentiation (**d**). In (**c**,**d**), each UMAP plot names the cluster(s) expressing the indicated gene. EC, endothelial cells; li. EryP, liver primitive erythrocytes; Mkb, megakaryoblasts; NA, not assigned; STC/SC, septum transversum cells/hepatic stellate cells.

**Figure 6 jdb-11-00015-f006:**
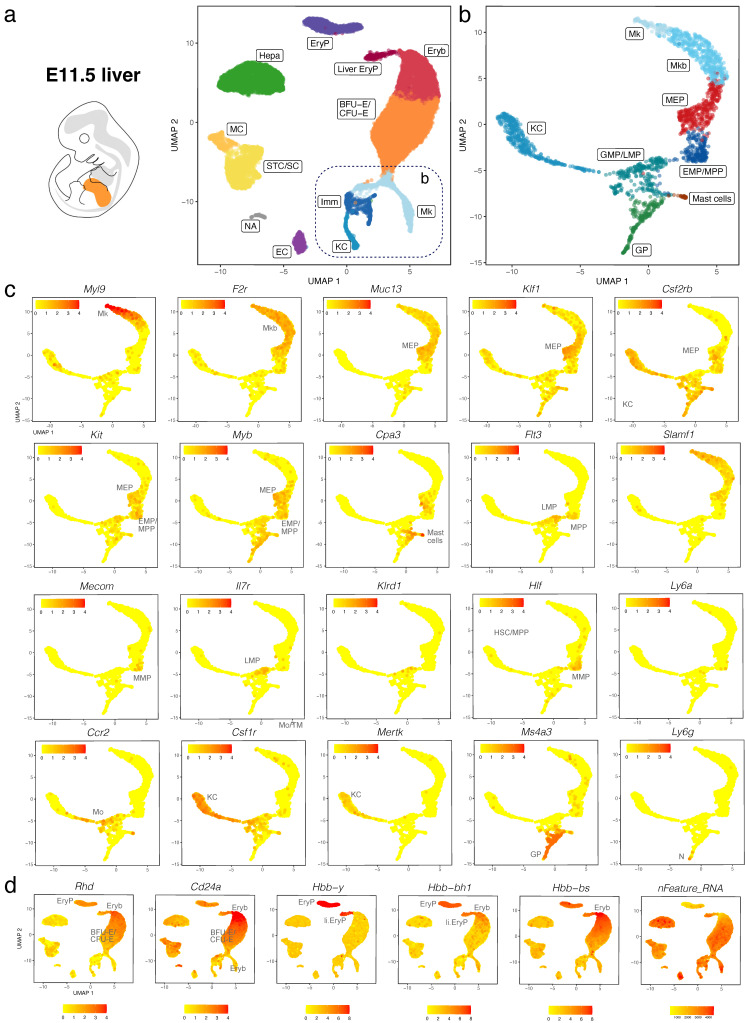
scRNA-seq of the E11.5 mouse liver. (**a**,**b**) The UMAP plot in (**a**) visualises clusters of distinct cell types in the total dataset; the box indicates the Mk, KC and Imm clusters, which were subclustered in (**b**). (**c**,**d**) UMAP plots visualise expression of the indicated genes that serve as markers of distinct branches of haematopoietic cell (**c**) and erythroid differentiation (**d**). In (**c**,**d**),each UMAP plot names the cluster(s) expressing the indicated gene; MC, mesothelial cells.

**Figure 7 jdb-11-00015-f007:**
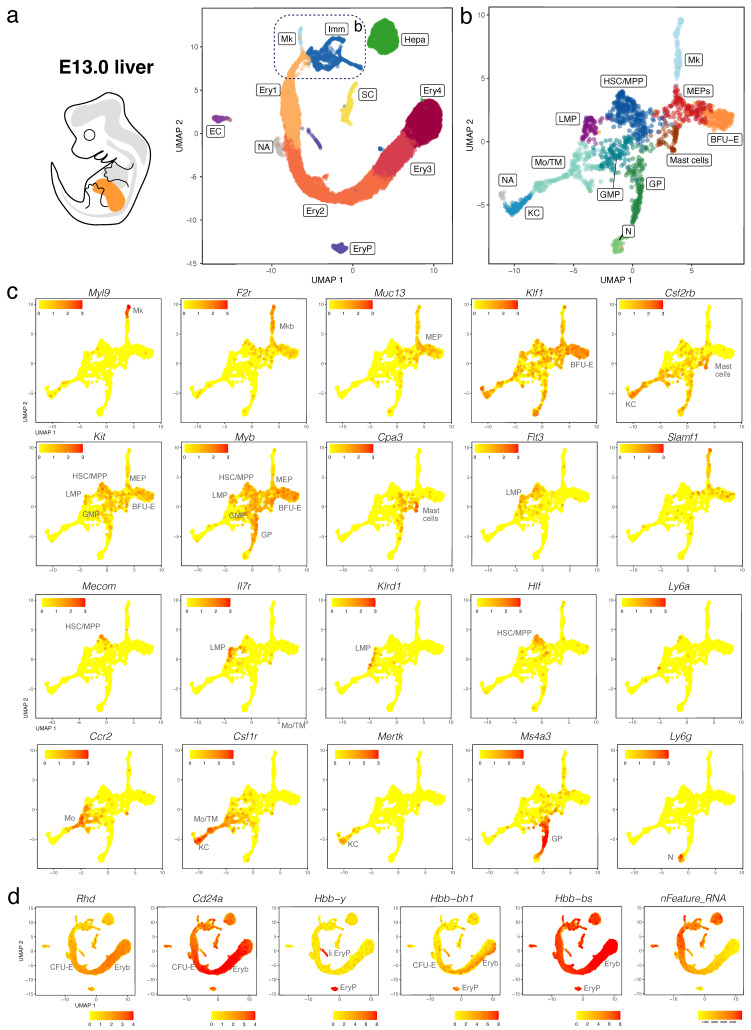
scRNA-seq of the E13.0 mouse liver. (**a**,**b**) The UMAP plot in (**a**) visualises clusters of distinct cell types in the total dataset; the box indicates the Mk, BFU-E and Imm clusters, which were subclustered in (**b**). (**c**,**d**) UMAP plots visualise expression of the indicated genes that serve as markers of distinct branches of haematopoietic cell (**c**) and erythroid differentiation (**d**). In (**c**,**d**), each UMAP plot names the cluster(s) expressing the indicated gene; SC, hepatic stellate cells.

## Data Availability

The E12.5 foetal liver scRNA-seq has been deposited in NCBI’s Gene Expression Omnibus (NCBI-GEO, GSE180050) and can be explored via the interactive website: https://ececcacci.shinyapps.io/LiverE12/ (accessed on 23 January 2023). Other publicly available datasets analysed in this study include: whole mouse embryo scRNA-seq at E8.5 (ArrayExpress, E-MTAB-6967); foetal liver scRNA-seq at E9.5, 10.5 and 11.5 (NCBI-GEO, GSE87038); foetal liver scRNA-seq at E11.0, 11.5 and 13.0 (CNCB-NGDC, CRA002445). All code used is available upon request.

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
