# Peer review of "A Refined Single Cell Landscape of Haematopoiesis in the Mouse Foetal Liver"

_jdb, 2023, doi:10.3390/jdb11020015_

Round 1

Reviewer 1 Report

This study focuses on defining the cell types within the murine E12.5 fetal liver, based on scRNAseq analysis of their own data and other published data. The hematopoietic-related data could be a useful resource for the field, but some issues should be addressed, as detailed below.

1. The Introduction details the steps of fetal liver development, and it could be useful to have an accompanying diagram (even as supplemental data) for readers to follow, especially those less familiar with the process. Or shorten it overall and highlight only the events relevant to the rest of the paper.

2. Details in the Introduction sometimes move back and forth between mouse and human. This is a little confusing – perhaps discuss human first, for context, and then events/cell types in the mouse relevant to what is under investigation. The chronology of developmental and postnatal events, in both human and mouse, is also not consistent and could be clearer.

3. In the analysis of the scRNAseq data, what was the contribution of each analyzed sample (2 technical reps each of 2 biological samples) to the total number of cells (10,537) that were analyzed?

4. It is not clear how SCTransform was used to integrate the data. Were technical replicates also integrated through SCTransform? Were there differences in the batch effects between samples and replicates?

5. There was a very low yield of endothelial cells and hepatocytes in the scRNAseq data, relative to erythrocyte clusters and other hematopoietic cells. Other vascular and hepatic cell types are present at this stage of development and the liver is highly vascularized; thus, why were they not well represented? Was this a tissue digestion or sample-processing issue? Please explain in the Results section.

6. It is not clear why Sox18 was used as an endothelial cell marker, especially since it is expressed by endodermal cells that form the liver. Are there other “classical” markers of endothelial cells expressed in this cluster?

7. The cell cycle scoring in Fig 3D and 4B are interesting, but it is not clear how accurate it is for the clusters that have very small cell numbers (e.g. Mo, Mkb, Mkc, SC). Also, cell cycle scoring was used to regress their data during their scaling step, so the authors should explain why their cell cycle scoring regression of gene expression does not affect their cell cycle state interpretations.

8. On line 154, it states “…the Sand G2M scores…” and this should be “…the S and G2M scores”.

9. The scRNAseq analysis of publicly available datasets was not detailed. If the data are from different platforms/library prep/batches, how was it processed and controlled for? Were the sequencing data generated by these investigators included along with these other data in a common pipeline for analysis or were comparisons made from only these other data.

10. In the Results and Discussion sections, statements like these (“This subset likely represents liver monocyte-derived tissue macrophages that will gradually replace the initial pool of yolk sac-derived Kupffer cells during late gestation (Ginhoux and Guilliams, 2016).” and “Prior to E12.5, foetal liver haematopoiesis is therefore predominantly driven by transient definitive progenitors first appearing in the mouse liver at E10.5, rather than definitive progenitors.” and “Cells in this cluster are likely hepatoblasts undergoing hepatocyte differentiation.”) should be avoided because there is no experimental evidence provided to support them.

11. Results Section 3.8, as well as related figure and Discussion, should be excluded. The data do not add to the paper and the analysis is based on very low cell numbers. Also, statements like this (“These findings suggest that at E12.5 LSECs were composed by endothelial cells that either began acquiring a central vein zonation phenotype (LSEC1) or lacked a zonation-specific phenotype (LSEC2).” are not experimentally supported and can be misleading, so should be avoided.

Reviewer 2 Report

The present work described the generation of a scRNA-seq dataset from a mouse foetal liver, which was obtained without cell preselection and includes a larger number of cells than prior datasets at this stage. I have the following concern related to this manuscript.

QC information is missing, please provide the QC plots and parameters used in detail.

Information related to cluster resolution is missing.

Subpopulations of cells identified Imm, Mk, and Ery1 clusters were not explained well, and some of the abbreviations of cell types were not described (Fig 2). It is challenging to correlate the data in Figures 2 & 3 and sections 3.2 and 3.3. It would be easier if both were merged and explained more thoroughly.

Why were only 1000 variable features in the SCTransform function used?
